# Ultrasound-Mediated Blood-Brain Barrier Opening Improves Whole Brain Gene Delivery in Mice

**DOI:** 10.3390/pharmaceutics13081245

**Published:** 2021-08-12

**Authors:** Marie-Solenne Felix, Emilie Borloz, Khaled Metwally, Ambre Dauba, Benoit Larrat, Valerie Matagne, Yann Ehinger, Laurent Villard, Anthony Novell, Serge Mensah, Jean-Christophe Roux

**Affiliations:** 1Aix Marseille Univ., INSERM, MMG, U1251, Faculté de Médecine Timone, 13385 Marseille, France; marie-solenne.FELIX@univ-amu.fr (M.-S.F.); emilie.BORLOZ@univ-amu.fr (E.B.); Valerie.MATAGNE@univ-amu.fr (V.M.); Yann.Ehinger@ucsf.edu (Y.E.); Laurent.VILLARD@univ-amu.fr (L.V.); 2Aix Marseille Univ., CNRS, Centrale Marseille, LMA UMR 7031, 13013 Marseille, France; metwally@lma.cnrs-mrs.fr (K.M.); mensah@lma.cnrs-mrs.fr (S.M.); 3Université Paris-Saclay, CEA, CNRS, Inserm, BioMaps, Service Hospitalier Frédéric Joliot, 91401 Orsay, France; ambre.dauba@universite-paris-saclay.fr (A.D.); anthony.novell@universite-paris-saclay.fr (A.N.); 4Université Paris-Saclay, CNRS, CEA, DRF/JOLIOT/NEUROSPIN/BAOBAB, 91191 Gif-sur-Yvette, France; benoit.larrat@cea.fr

**Keywords:** gene therapy, AAV9, focused ultrasound, blood-brain barrier, microbubbles

## Abstract

Gene therapy represents a powerful therapeutic tool to treat diseased tissues and provide a durable and effective correction. The central nervous system (CNS) is the target of many gene therapy protocols, but its high complexity makes it one of the most difficult organs to reach, in part due to the blood-brain barrier that protects it from external threats. Focused ultrasound (FUS) coupled with microbubbles appears as a technological breakthrough to deliver therapeutic agents into the CNS. While most studies focus on a specific targeted area of the brain, the present work proposes to permeabilize the entire brain for gene therapy in several pathologies. Our results show that, after i.v. administration and FUS sonication in a raster scan manner, a self-complementary AAV9-CMV-GFP vector strongly and safely infected the whole brain of mice. An increase in vector DNA (19.8 times), GFP mRNA (16.4 times), and GFP protein levels (17.4 times) was measured in whole brain extracts of FUS-treated GFP injected mice compared to non-FUS GFP injected mice. In addition to this increase in GFP levels, on average, a 7.3-fold increase of infected cells in the cortex, hippocampus, and striatum was observed. No side effects were detected in the brain of treated mice. The combining of FUS and AAV-based gene delivery represents a significant improvement in the treatment of neurological genetic diseases.

## 1. Introduction

One of the biggest challenges in developing therapeutic agents for the treatment of neurological disorders is the impermeability of the blood-brain barrier (BBB). It has been noted that 98% of drugs fail in clinical trials due to the difficulty to cross the BBB [1]. The BBB is a highly selectively permeable cellular phospholipid protein bilayer barrier that separates the circulating blood from the brain. The BBB is composed of capillary endothelial cells, connected by tight junctions [2]. The BBB limits the entry of proteins larger than 0.5 kDa, requiring active mechanisms for larger molecules to enter the central nervous system (CNS).

Gene therapy approaches represent a tremendous technological advance opening doors to the treatment of many genetic or traumatic diseases [3]. The CNS is among the most targeted organs but its infection by therapeutic viral vectors is hampered by the BBB protecting it and thus limiting neuronal infection rates [4]. Since several adeno-associated viruses (AAV) serotypes were found to have some ability to enter the CNS, AAVs are widely used vectors in gene therapy protocols targeting the CNS [5]. Preclinical proofs of concept have been obtained, particularly with AAV9, allowing their potential use for clinical applications [6,7]. However, the delivery of AAV vectors in the CNS in a non-invasive way, i.e., through the bloodstream, results in poor infection rate [3]. Therefore, when large brain areas or the entire CNS need to be treated, the most efficient method involves multiple intracerebral injections, which are not without risk to the patient [4,8,9,10]. To date, only lethal pathologies or very severely affected patients have been chosen for this approach [7].

Focused ultrasound (FUS) coupled with the use of circulating microbubbles (MB) represents an innovative technology for temporarily disrupting the BBB [11,12]. Contrast agent MB are efficient sensitizers of mechanical stress under bulk-pressures of varying amplitude [13]. These gas-filled 1–5 μm MB can repeatedly expand and compress with no risk of collapsing when they are exposed to low enough acoustic pressures. This oscillation regime, also called stable cavitation oscillating mode, leads to tight junctions opening via mechanical stress on the endothelium. These rapid expansions and contractions can also create micro-streams able to disrupt the endothelial lining by shear stress [14]. When located inside the vessels, stable cavitation can be induced by applying transcranial ultrasound with moderate power. This last strategy is promising since it allows the localized permeabilization of the BBB by combining FUS with intravenously injected MB [11].

Many groups have started to use this tool to deliver AAVs into specific brain regions [15,16,17]. A different protocol may be necessary when gene delivery is needed over a large part of the brain. For example, ischemia, Amyotrophic Lateral Sclerosis, Alzheimer’s disease, Huntington’s disease, Fragile X, Down Syndrome, or Rett syndrome are pathologies affecting the whole brain or at least a broad brain region. This implies the need for a global disruption of the BBB [18,19]. To our knowledge, no study has investigated the use of FUS to deliver AAVs to the whole brain. Therefore, the aim of this study was to develop a whole-brain BBB-opening protocol, both safe and effective, using a FUS BBB-opening device assisted by automated CNS scanning system.

## 2. Materials and Methods

### 2.1. Animals

Thirty wild-type (WT) C57BL/6J 7-weeks old mice were studied, both male and female (mean weight: 15.5 ± 1.9 g), and split into several experimental groups (Table 1). WT C57BL/6J mice were purchased from Charles River and maintained on a pure C57BL/6J genetic background. Mice were housed under a 12:12 h light ⁄ dark cycle (lights on at 07:00) and given free access to ad libitum food and water. Experimental protocols were approved by ethics committee CE14 of the Faculty of Medicine of Marseille (APAFIS #220319). All experiments were conducted in compliance with the European guidelines for the care and use of laboratory animals (EU directive 2010/63/EU), and the guide for the care and use of the laboratory animals of the French national institute for science and health (Inserm).

### 2.2. Focused Ultrasound Device

The focused ultrasound (FUS) blood-brain barrier (BBB) opening device can be decomposed into two parts: an active FUS probe and a scanning platform.

The FUS probe consisted of a concave spherically shaped piezocomposite transducer used to transmit a sinusoidal wave at 1.5 MHz (diameter 25 ± 0.1 mm, focal depth 20 ± 2 mm, Imasonic, Voray-sur-l’Ognon, France). This transducer has a hole in its center to house a cavitation detector that was not used in the present study. The measured focal volume of the FUS transducer was 1 × 1 × 6 mm^3^ at −6 dB.

In a calibration step, the output pressure was measured in a degassed water tank, using a 0.5 mm needle hydrophone (Precision Acoustics, Dorchester, UK) mounted on a positioning stage. The transducer was driven by a built-in signal generator connected to a 50 W power amplifier (Image Guided Therapy, Pessac, France). The signal acquired from the hydrophone was sampled by the oscilloscope (Picoscope 5243A, Pico Technology, St Neots, UK) and transferred to a personal computer for analysis. After scanning the whole focal spot, the calibrated hydrophone was placed in the center of the focal area to record the relationship between the electrical input and acoustic pressure.

The FUS transducer was then coupled to the head of the animal via a water balloon filled with deionized and degassed water to outreach 7 to 10 mm from the surface of the transducer. For that purpose, a thin and transparent latex-based membrane was fixed to the transducer and a water filling circuit was connected in order to fill or empty the balloon. This balloon-size flexibility permits a scanning up to 13 ± 2 mm in depth within the head depending on the targeted area. After being shaved (depilatory cream), the head of the mouse is covered with degassed echography gel (70% *v*/*v* in water) to ensure a good coupling and to avoid air bubbles retention.

For the protocol used, no feedback based on cavitation activity was applied to control the electrical power sent to the transducer during the BBB opening session. A peak negative pressure (PNP) of 0.65 MPa at the focus in a water bath was applied. This referenced pressure has been adjusted in order to reach desired in situ pressure behind the skull. To do so, mean acoustic transmission through various mouse skulls has been measured as detailed below.

An in vivo platform (Image Guided Therapy, Pessac, France) comprises a stereotaxic frame, a transducer holder mounted on fully programmable 3D scanning stages and programmable ultrasound single-channel amplifier and single channel receiver (for future cavitation detection).

### 2.3. Ultrasound Transmission through the Skull

In all, 12 young WT mice (day 10 to day 45, body mass from 5 g to 20 g, 5~10 g: *n* = 3, 10~15 g: *n* = 3, 15~20 g: *n* = 6) were anesthetized with euthasol (10 mg/mL) i.p. (intraperitoneal) injection then perfused with PBS only. Cranial bones were dissected and rinsed in PBS before being preserved in PBS azide (0.02% S2002 Sigma-Aldrich, St Louis, MO, USA) and stored at 4 °C until further use. Few hours before ultrasound measurements, skulls were immersed into deionized water and degassed for 6 h using a chamber connected to a vacuum pump (Rocker 400, Dutscher, Bernolsheim, France) to remove any air pockets trapped into the bones. Transmission coefficients of the skulls were measured by the through-transmission substitution technique [20] at 1.5 MHz. Acoustical measurements were performed before and after insertion of the skull (parietal bones) in a degassed-water tank maintained at room temperature. Ultrasound waves were generated from the single-element transducer (Imasonic, Voray-sur-l’Ognon, France) focused at 20 mm connected to the therapeutic platform (50 W, Image Guided Therapy). For each sample, a 20-cycle excitation tone burst was transmitted at a PNP of 0.65 MPa at the focus (in water). The delivered excitation pulses were measured using a capsule hydrophone (HGL-200, ONDA, Sunnyvale, CA, USA) mounted on a three-axis motorized stage (10 µm accuracy, Image Guided Therapy) and placed at the focal distance of the transducer. Both the transducer and the skull sample were mounted on a holder to be static during the experiment. The pulses received by the hydrophone were captured with the oscilloscope (MSO54, Tektronix, Beaverton, OR, USA), and data was transferred to MATLAB (R2019b, The Mathworks, Natick, MA, USA) for offline analysis. Ultrasound transmission (in %) through the skull was calculated by determining the average peak amplitude and normalizing it by the reference (water only) acquired before the skull insertion. For each sample (*n* = 12), experiments were repeated 3 times at different locations of the parietal skull.

### 2.4. Viral Vector Preparation and In Vivo AAV Injections

Self-complementary recombinant pseudotyped AAV2/9 vectors were produced by the Vector Core at the University Hospital of Nantes based on the protocol of Ayuso et al. [21]. The scAAV9 vectors, that were used in this study contained a cassette expressing GFP (AAV9-GFP) under the control of the CMV promoter, were previously characterized [22].

The mice were anesthetized using isoflurane (IsoVet, Piramal Healthcare, Northumberland, UK) induction in a cage at 4% following by anesthesia under a mask at 2% and injected in retro-orbital sinus with AAV9 vectors at a dose of the 1.1 × 10^11^ vg/mouse using a 29-gauge needle.

### 2.5. FUS-Induced BBB Disruption

Our system allows performing a fast and continuous raster scan (10 mm·s^−1^) during the FUS-emissions. Specifically, considering a row spacing of 1.2 mm, a square surface (6 mm × 6 mm) that covers the main part of the brain is scanned iteratively (Figure 1) 30 times every 5 s for a total duration of 150 s, while as the ultrasounds are transmitted following a raster scan with a pulse duration of 100 ms and a duty cycle of 99%.

During the anesthesia, each animal was also injected into retro-orbital sinus with 100 µL of sulphur hexafluoride microbubbles (8 µL/mL, mean diameter 2.5 µm, SonoVue, Bracco, Milano, Italy) using a 29-gauge needle. We are aware that using needles smaller than 23-gauge could result in a loss of MB at injection [23]. Therefore, we injected an excess of MB (4.5 µg) compared to clinical dose.

For half of the mice (*n* = 15), the injection was followed by the transcranial application of continuous FUS along the 2D square-path trajectory (speed 10 mm·s^−1^) to disrupt the BBB over the whole brain volume (FUS-treated mice). For the other half (control mice, *n* = 15), the injection of microbubbles was not followed by FUS application. The calibrated sonication parameters allowing BBB opening were fixed in order to deliver an in situ PNP of 0.57 MPa at the focus point.

### 2.6. Tissue Collection

Animals were sacrificed either 24 h or 1 month after FUS treatment, for safety and efficacy studies respectively. Mice were euthanized with an overdose of euthasol (pentobarbital at 362.9 mg/mL) diluted to 10 mg/mL in PBS. Tissue collection was different depending on the following experiments:(1)For H&E staining, TUNEL staining assay, and cleaved caspase-3 immunohistofluorescence, mice were transcardially perfused with 0.01 M PBS followed by 4% paraformaldehyde (PFA) in phosphate buffer, then brains were cryopreserved in 20% sucrose for 2 days and frozen in dry ice. Tissues were embedded in Tissue-Tek CRYO-OCT compound (Fisher Scientific, Illkirch, France), coronally sectioned from bregma 1.34 mm to bregma −3.28 mm into 20 or 40 μm sections using a Leica VT1200s cryostat (Leica Biosystems, Nanterre, France), mounted on SuperFrost Plus slides (Fisher Scientific, Illkirch, France), and stored at −80 °C until assay.(2)For Iba1, Olig2 and GFAP immunohistofluorescence and GFP immunohistochemistry, mice were transcardially perfused with 0.01 M PBS followed by 4% paraformaldehyde (PFA) in phosphate buffer, then brains were kept at 4 °C in PBS azide (0.02% S2002 Sigma-Aldrich, St Louis, MO, USA), cut in 100 µm sections using a vibratome (Leica VT1200s, Leica, Nanterre, France), and collected serially. Floating sections were kept at 4 °C in PBS azide (0.02% S2002 Sigma-Aldrich, St Louis, MO, USA) until assay.(3)For GFP quantification (DNA, mRNA, and protein preparations), brains were rapidly dissected and frozen in dry ice.

### 2.7. H&E Staining

Frozen slides (sections of 40 µm) were left at room temperature for an hour and then rehydrated by successive incubation in 95% and 70% ethanol, and then water for 2 min before incubation in Vector hematoxylin solution (Eurobio Scientific, Les Ulis, France) for 60 sec. After rinsing in water, the slides were incubated for 30 sec in Eosin Y solution (Sigma-Aldrich, St Louis, MO, USA, # 318906, diluted 1:10 in 0.25% acetic acid and 70% ethanol) and then dehydrated by successive 2 min incubation in ethanol 70%, 95%, and 100%. Sections were then cleared in xylene (VWR, Fontenay-sous-bois, France) for 2 min and mounted in DPX mounting media (Fisher Scientific, Illkirch, France). Pictures were taken in the cortex, the corpus callosum, the hippocampus, the basal ganglia, the midbrain, the thalamus, and the hypothalamus on 10× magnification (aperture 0.30 HCX PL FLUOTAR) on a Leica DM 5000B microscope equipped with a camera Leica (DFC 300 FX) and the magnification. Images were collected using the software Las V4.9 (2017, Leica, Wetzlar, Germany). Exposition times, image processing, and merging were performed using the same parameter within each experiment.

### 2.8. Cleaved Caspase-3 Immunohistofluorescence

Liver sections were used as a positive control for cleaved caspase-3 staining. Frozen slides of brain and liver (sections of 40 µm) were left at room temperature for 20 min and then were washed in phosphate-buffered saline 1× (PBS) buffer (from PBS 10×, Gibco, Life Technologies, Waltham, MA, USA) 4 times (10 min), incubated in blocking solution (3% normal donkey serum (Jackson ImmunoResearch Europe, Ltd., Suffolk, UK), 0.3% Triton X-100, PBS 1×) for 1 h and transferred in rabbit anti-cleaved caspase 3 antibody (#9661, 1:400, CST) diluted in blocking solution or in blocking solution only (NO AB control sections) overnight at 4 °C. After washing the sections 4 times for 10 min, they were incubated for 2 h at room temperature with the secondary antibody (donkey anti-rabbit Alexa 596, 1:400, #R37117) diluted in blocking solution and then washed 4 more times in PBS. Nuclei were stained by 5 min incubation with DAPI (0.4 µg/mL) followed by 2 rinses in PBS. Sections were then coverslipped with Shandon Immu-Mount (Fisher Scientific, Illkirch, France). The immunolabeled slices were digitized and recorded using an Apotome Axioimager 2 (Carl Zeiss, Munich, Germany). Images were collected using the software Zen 2.3 (2016, Carl Zeiss, Munich, Germany). Exposition times, image processing, and merging were performed using the same parameter within each experiment.

### 2.9. TUNEL Staining Assay

Apoptosis in the brain was measured in vivo with a TUNEL staining assay. Frozen slides (sections of 20 µm) were left at room temperature for 20 min and then TUNEL assays were performed by using the Click-iT™ Plus TUNEL Assay (ThermoFisher Scientific, Illkirch, France) for in situ apoptosis detection with Alexa 488 Fluor™ dye according to the manufacturer’s protocol. Nuclei were stained by 5 min incubation with DAPI (0.4 µg/mL) followed by 2 rinses in PBS 1× for 5 min. Sections were then air-dried for 30 min and coverslipped with Shandon Immu-Mount (Fisher Scientific, Illkirch, France). Pictures were taken in the cortex, the corpus callosum, the hippocampus, the basal ganglia, the midbrain, the thalamus, and the hypothalamus on 10× magnification (aperture 0.30 HCX PL FLUOTAR) on a Leica DM 5000B microscope equipped with a camera Leica (DFC 300 FX) and the magnification. Images were collected using the software Las V4.9 (2017, Leica, Wetzlar, Germany). Exposition times, image processing, and merging were performed using the same parameter within each experiment.

### 2.10. Iba1, GFAP and Olig2 Immunohistofluorescence

Immunohistofluorescence was performed on free-floating sections (100-μm thick). Sections were washed in phosphate-buffered saline (PBS) and incubated for 30 min with 5% diluted normal donkey serum (Jackson ImmunoResearch Europe, Ltd., Suffolk, UK) and 0.5% Triton X-100; then incubated overnight at room temperature in the same solution with the primary antibodies rabbit anti-Iba1 (wako 019-19741), mouse anti-GFAP (cell signaling 3670), and a rabbit anti-Olig2 (Merck AB9610) diluted 1/500. After washing the sections 4 times for 10 min, they were incubated for 1 h at room temperature with the secondary antibodies donkey anti-rabbit 488 (Alexa FLuor Invitrogen A32790) and donkey anti-mouse 555 (Alexa FLuor Life A31570) diluted in blocking solution and then washed 4 more times in PBS. Nuclei were stained by 5 min incubation with DAPI (0.4 µg/mL) followed by 2 rinses in PBS. Sections were then coverslipped with Shandon Immu-Mount (ThermoFisher Scientific, Illkirch, France). The immunolabeled slices were digitized and recorded using an Apotome Axioimager 2 (Carl Zeiss, Munich, Germany). Images were collected using the software Zen 2.3 (2016, Carl Zeiss, Munich, Germany). Exposition times, image processing, and merging were performed using the same parameter within each experiment.

For the GFAP, Iba1 and Olig2 immunoquantification we used the Fiji software. Images were converted to 8-bit format, then thresholded, and a region of interest (ROI) was drawn on the image according to the brain structures in the Allen Mouse Brain Atlas (2004). The number of particles in the ROI was then counted with the Fiji particle analyzing tool. All slides were analyzed with the experimenter blind for the treatment (FUS or not) using the same level of detection. Three animals were counted for control condition and 3 for FUS-treated condition. Several slices were counted for one animal. This protocol has already been used [22].

### 2.11. GFP Immunohistochemistry

Immunohistochemistry was performed on free-floating sections (100-μm thick). Sections were washed in phosphate-buffered saline (PBS) and incubated for 30 min with 5% diluted normal donkey serum (Jackson ImmunoResearch Europe, Ltd., Suffolk, UK) and 0.5% Triton X-100; then incubated overnight at room temperature in the same solution with the primary antibody goat-anti GFP (#ab6673, 1:500; Abcam, Cambridge, UK). Secondary amplification and detection were performed using the VectaStain ABC Elite Kit (#PK-6101; Vector Labs, Eurobio, Courtaboeuf, France) and the Vector VIP Peroxidase (HRP) Substrate Kit (#SK-4600) that were used according to the manufacturer recommendation. Finally, sections were rinsed in tap water and mounted. Using the Fiji software, images were converted to 8-bit format, then thresholded, and a region of interest (ROI) was drawn on the image according to the brain structures in the Allen Mouse Brain Atlas (2004). The number of particles in the ROI was then counted with the Fiji particle analyzing tool. All slides were analyzed with the experimenter blind for the treatment. Six animals were counted for control condition and 6 for FUS-treated condition. Several slices were counted for one animal.

### 2.12. DNA Extraction and Quantitative Real-Time qPCR

Total DNA was isolated from brain samples using the DNeasy Tissue and Blood kit (Qiagen, Courtaboeuf Cedex, France) according to the manufacturer’s instruction. DNA concentration, quality, and purity were verified by electrophoretic trace (NanoDrop, ThermoFisher Scientific, Illkirch, France).

Quantification of *AAV-GFP* DNA (forward primer 5′-GACGACGGCAACTACAAGAC-3′, reverse primer 5′-TCCTTGAAGTCGATGCCCTT-3′) was made by quantitative real-time PCR using the QuantStudio^TM^ 5 system (ThermoFisher Scientific, Illkirch, France). Each reaction was performed in a 20 µL reaction containing 2 µL of total DNA (50 ng/µL), 0.2 µM of each primer, and 10 µL of the SYBR Green I Master kit (Roche LifeScience, Penzberg, Germany). Triplicates were run for each sample and the *Adora2b* gene (forward primer 5′-CCCAAGTGGGTGATGAATGT-3′, reverse primer 5′-GGGGTTGACAACTGAATTGG-3′) was used as an internal control. Gene quantification was obtained by the ΔΔCT relative quantification method [24].

### 2.13. RNA Extraction, Reverse Transcription, and Quantitative Real-Time qPCR

Total RNA was isolated from brain samples using the Illustra RNAspin^TM^mini kit (GE Healthcare Life Sciences) according to the manufacturer’s instructions. RNA concentration, quality, and purity were verified by electrophoretic trace (NanoDrop, ThermoFisher Scientific, Illkirch, France).

One microgram of total RNA was retro-transcribed using the Superscript IV enzyme according to the manufacturer’s instructions (Invitrogen, Life Technologies, Illkirch, France).

Quantitative assessment of *GFP* mRNA expression (forward primer 5′-TCCTTGAAGTCGATGCCCTT-3′, reverse primer 5′-GACGACGGCAACTACAAGAC-3′) was verified by quantitative real-time PCR using the QuantStudio^TM^ 5 system (Thermo Fisher Scientific, Illkirch, France). Each reaction was performed in a 20 µL reaction containing 2 µL of cDNA (diluted at 1:20), 0.2 µM of each primer, and 10 µL of the SYBR Green I Master kit (Roche LifeScience, Penzberg, Germany). Triplicates were run for each sample and the *Hmbs* gene (NM_001110251.1, forward primer 5′-TCCCTGAAGGATGTGCCTAC-3′, reverse primer 5′-CACAAGGGTTTTCCCGTTT-3′) was used as an internal control. Gene expression was obtained by the ΔΔCT relative quantification method [24].

### 2.14. Western Blotting

The tissue samples were homogenized in lysis buffer (Tissue Extraction Reagent I (#FNN0071, Invitrogen^TM^, Waltham, MA, USA), 1X Pierce Halt^TM^ Protease Inhibitor Cocktail and 1X Pierce Halt^TM^ Phosphatase Inhibitor Cocktail) by grinding in liquid nitrogen and passing through a 23-gauge needle. The protein extracts were then centrifuged for 15 min at maximum speed to pellet tissue debris and the protein concentration was determined using the Pierce BCA Protein assay (ThermoFisher Scientific, Illkrich, France). One hundred and fifty µg protein per lane were separated on a 4–20% Mini-PROTEAN^®^ TGX™ Precast Protein Gel (Bio-Rad, Marnes-la-Coquette, France) and transferred on nitrocellulose membrane using the Trans-Blot^®^ Turbo^TM^ blotting system (Bio-Rad, Marnes-la-Coquette, France). The membrane was blocked in Immobilon^®^ Block—FL (Millipore, #WBAVDFL01, Burlington, MA, USA) at room temperature for 1 h and incubated 2 h in primary antibody (Goat anti-GFP (#ab6673; 1:10,000, Abcam, Cambridge, UK) and mouse anti-GAPDH (#ab181602; 1:10,000, Abcam, Cambridge, UK) in TBS 0.1% Tween-20 with mild agitation at room temperature. The membranes were washed 3 times in TBST for 5 min and incubated with an anti-mouse IRDye 800CW and anti-goat IRDye 680RD secondary antibody (LI-COR Biotechnology) diluted at 1:10,000 in 20% blocking solution at room temperature for 1 h. After rinsing the membranes in TBST, the blots were visualized on a ChemiDoc MP imaging system (Bio-Rad, Marnes-la-Coquette, France).

Quantitation analysis (measurement of integrated pixel volume) was performed using Fiji (1.53c, 2020, ImageJ, NIH, Bethesda, MD, USA).

### 2.15. Statistical Analysis

All statistical analyses were performed using GraphPad Prism v.9 for Windows/macOS (GraphPad Software, La Jolla, CA, USA). Comparison between 2 groups was analyzed by a Mann–Whitney rank-sum test. Data are expressed as means ± SEM and *p* < 0.05 was considered significant for all tests. Cohen’s d effect size was calculated as follows:(1)$$\frac{mean\;FUS\;treated\;mice-mean\;control\;mice}{\sqrt{\frac{mean\;FUS\;treated\;mice^{2}}{n\;FUS\;treated\;mice}+\frac{mean\;control\;mice^{2}}{n\;controle\;mice}}}$$

## 3. Results

### 3.1. Ultrasound Transmission through the Skull

For young mice, the transmission loss through the skull can vary from 0.5% to 30% (Figure 2) and is correlated to the animal body mass (*R*^2^ = 0.92). Considering body mass (BM in g) from 5 to 20 g, the transmission loss in acoustic pressure (TL in %) can be expressed using the following formula:TL = 0.33 *e* ^0.23×BM^(2)

### 3.2. Short-Term Safety after FUS-Treatment

Apoptosis and hemorrhages were investigated 24 h after AAV9-GFP and microbubbles (MB) injection and FUS application (Figure 3) to a brain area that included the cortex, the corpus callosum, the hippocampus, the basal ganglia, the midbrain, the thalamus, and the hypothalamus. Contrary to the positive control that was submitted to sublethal doses of FUS (Figure 3C), hematoxylin-eosin staining showed neither red blood cell infiltration nor gross histological abnormality in the brain of FUS-treated mice (Figure 3B) or non-FUS mice (Figure 3A). Apoptosis in the brain was studied by cleaved-caspase 3 immuno-fluorescent staining (Figure 3D–F) and TUNEL assay (Figure 3G–I). Cleaved-caspase 3 and TUNEL stainings were absent both in non-FUS (Figure 3D,G) and FUS-treated mice (Figure 3E,H). Together, these results confirm the absence of apoptosis in FUS-treated mice.

### 3.3. FUS-Treatment Homogeneously Increased AAV9 Vector Transduction in the Brain

Efficacy studies showed that FUS-treatment homogeneously increased AAV9 transduction and expression in the brain (Figure 4A–C). *GFP* DNA (Figure 4A), *mRNA* (Figure 4B), and protein (Figure 4C) levels were increased in the brain of FUS-treated mice compared to control mice (*n* = 3 in each group, 19.8-, 16.4-, and 17.4-fold increase, respectively). The number of GFP-positive cells was significantly increased in the FUS-treated group compared to control in several brain structures, including the striatum, cortex, and hippocampus. A substantial 9.9-fold increase of GFP positive cells was found in the striatum while it was 5.1- and 6.8-fold increases in the cortex and the hippocampus, respectively (Figure 4F–H). Moreover, by comparing cell counts between the two hemispheres, no statistical differences were found with only 16% difference (right versus left) in the control striatum and 0% in the FUS-treated striatum. Similarly, 4% of hemisphere difference was observed in the control cortex and 1% in the FUS-treated cortex, 5% in the control hippocampus and 2% in the FUS-treated hippocampus. Together these results show the efficacy of the FUS technique and the homogeneity of the infection in the brain.

Interestingly, while the brainstem region outside the FUS targeted area, a 4.6-fold increase in positive GFP cells was measured in the brainstem of FUS treated-animals in comparison with control condition (Appendix A).

### 3.4. Long-Term Safety after FUS Treatment

To determine whether FUS treatment induced neuroinflammation, microglial activation, white matter damage, and astroglial reactivity were measured 1 month after MB injection and FUS application (Figure 5) using known markers of reactive glia (GFAP, [25]), white matter (Olig2, [26,27]), and microglia (Iba1, [28]). Images were taken in the hippocampus and the cortex that were used as representative brain areas. In FUS-treated animals, as well as in non-FUS treated animals, no significant astrogliosis or reactive astrocytes were observed in the cortex (Figure 5A,B,I,J, Appendix A) or the hippocampus (Figure 5C–K, Appendix A). No significant microglia activation was observed in the cortex (Figure 5F, Appendix A) or the hippocampus (Figure 5H, Appendix A) of the FUS-treated animals when compared with non-FUS treated cortex (Figure 5E, Appendix A) or hippocampus (Figure 5G, Appendix A). Together, these results support the long-term safety of this technique as shown by the global absence of inflammatory reaction and no obvious abnormalities in the microglial cells number 1 month after FUS treatment.

## 4. Discussion

To date, all human CNS gene therapy studies have required direct surgical delivery due to the size of the viral particles and the presence of an intact BBB, which prevents the efficient transfer of viruses from the blood to the brain [10]. Increasing vector dosage to compensate for poor permeability is not an option since such a strategy can lead to severe side effects with the massive transduction of AAV in several off-target organs such as the liver [29]. In rodents, the use of FUS coupled with MB injection to increase the efficacy of gene therapy in the brain has mainly been used to target specific brain areas [30]. Here, we used an automated system to scan a large portion of the upper skull of young adult mice in order to target the whole brain [31,32], and our results showed that this protocol allowed efficient and safe BBB disruption in rodents. Concretely, the focal spot of the circular transducer had a 1 mm width, and it was moving over the head of the animal at a velocity of 10 mm·s^−1^. It means that each spot along the trajectory was insonified for 0.1 s. The raster scan was repeated 30 times every 5 s. Therefore, the effective duty cycle for a targeted area was in the order of 2%. Ultrasound transmission through the skull of mice (from 5 g to 20 g BW) was performed to determine the FUS transmission loss at 1.5 MHz and its correspondence with animal mass. Transmission loss can vary up to 30% and must be considered to obtain optimal and safe BBB disruption. Our results suggest that FUS procedure must be personalized to each animal. This is particularly necessary for animal models including young mice or pathology associated with growth retardation such as for Rett syndrome [33]. Transmission loss can easily be compensated by considering the animal body weight (Equation (2)) and adjusting the acoustic pressure accordingly. The relationship between body mass and transmission loss through the skull was also demonstrated in rats in a prior study [34]. Ultrasound beam deflection through the mouse skull is considered to be negligible at 1.5 MHz.

As previously shown [31], a continuous and generalized opening of BBB in the whole brain is safe and does not generate side effects. In line with previous work, our present results indicate that there are no obvious signs of side effects 24 h after FUS application and AAV injection. FUS used at high power levels can damage the microvasculature and allow paracellular extravasation of erythrocytes [35]. During the development of the protocol, we found that the use of a transcranial peak negative pressure (PNP) of 0.82 MPa in situ led to the death of a large proportion of mice (average weight: 16 g) and in all cases to important cerebral hemorrhages (Appendix A). For mice of mean weight of 15.6 g, the selected PNP value of 0.65 MPa in water (0.57 MPa in situ) did not lead to cell death, signs of inflammation, or lymphocyte infiltration as shown by hematoxylin-eosin staining and cell death/neuroinflammation marker staining (Figure 3 and Figure 5). FUS can also cause apoptosis [11,36]. In our study, no apoptotic cells were detected using cleaved caspase-3 staining or tunnel assay in any of the selected brain areas. Inflammatory response after FUS-induction was assessed by evaluating the proportion of microglia and by monitoring the shape of the labeled cells, as an indicator of their activation [37]. Indeed, the inflammatory response of the CNS is mediated by the activated microglia, resident immune cells, which normally respond to neuronal damage and remove the affected cells by phagocytosis [38]. No differences were observed between FUS and non-FUS animals in the brain areas evaluated in our study. Similarly, brains were stained with a reactive astrocyte marker in order to identify whether FUS could create astrogliosis [39]. Our results clearly showed an absence of glial inflammation induction in the brains of FUS-treated mice compared to the untreated ones. Although our safety results support the use of FUS as a whole brain transduction method, it does not guarantee that other biological parameters, not evaluated here, are not affected. For instance, a small inflammation or slight transcriptional deregulations could appear a few hours after the use of FUS [40,41].

Regarding the infection efficacy of scAAV9-CMV-GFP, our protocol shows a robust increase in vector transduction for all measured biological values. Indeed, the number of infected cells and the level of infection, indicated by the amount of vector, show a very clear added value of FUS. The number of infected cells increased by 6 to 10 times depending on the brain area and the amount of GFP-DNA increased by 19.8 times in the whole brain. The discrepancy between the increased number in infected cells and in the amount of vector DNA in the brain could be attributed to the sensitivity threshold of the GFP staining method and/or the fact that some cells have been infected with several AAVs simultaneously [42]. Interestingly, even if the central parts of the brain were more strongly infected after FUS treatment, a posterior part of the brain, such as the brainstem, also showed a significant increase in the number of GFP-positive cells (4.6-fold increase compared to non-FUS mice, Appendix A). The fact that the brainstem also benefited from FUS treatment is noteworthy as it is an important target in cardiorespiratory and autonomic phenotypes [43]. This broader infection and dissemination of AAVs [44] could be explained by their reaching the cerebrospinal fluid and therefore the brain tissue in the rostrocaudal axis and the spinal cord via the ventricles [45]. Ventricles are a known passageway to the brain and have successfully been used to deliver therapeutic agents such as antibiotics [46], as well as reporter or therapeutic transgenes using intraventricular AAV injections [45,47,48].

Another very strong point of our study comes from the homogeneity of the infection. Looking at the 3 main structures (striatum, hippocampus, and cortex), the added value of the FUS appears to be of the same magnitude. Comparison of the contralateral regions (left versus right hemispheres) of the brain reveals minimal differences in AAV infection ranging from 0 to 6%. A strong feature of our study is to have an increase of the mRNA and the protein levels (16.4 and 17.4 times, respectively). The substantial increase due to FUS should be considered especially when overexpression of a given transgene should be avoided. Indeed, some gene expression must be finely regulated. This is the case, for example, of the Mecp2 protein, which causes a pathology when the protein is defective, but also another neurological pathology when it is too abundant [49].

## 5. Conclusions

Altogether, our results indicate that the use of FUS technology can be generalized to the whole brain to favor a global and homogeneous gene delivery. The added value of FUS is strong and this tool makes it possible to reduce the vector dose used to target the brain and thus to limit the side effects. However, it will surely have to be adjusted in case the gene of interest requires a precise dosage. In summary and to the best of our knowledge, this technology is free of major side effects and represents a very promising approach to treat patients with neurological diseases affecting broad areas of the brain.

## Figures and Tables

**Figure 1 pharmaceutics-13-01245-f001:**
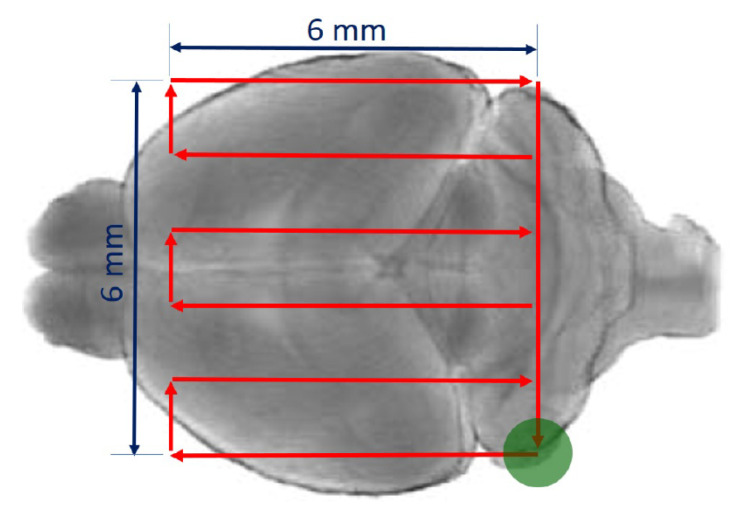
Scanning trajectory (in red) over the mouse brain.

**Figure 2 pharmaceutics-13-01245-f002:**
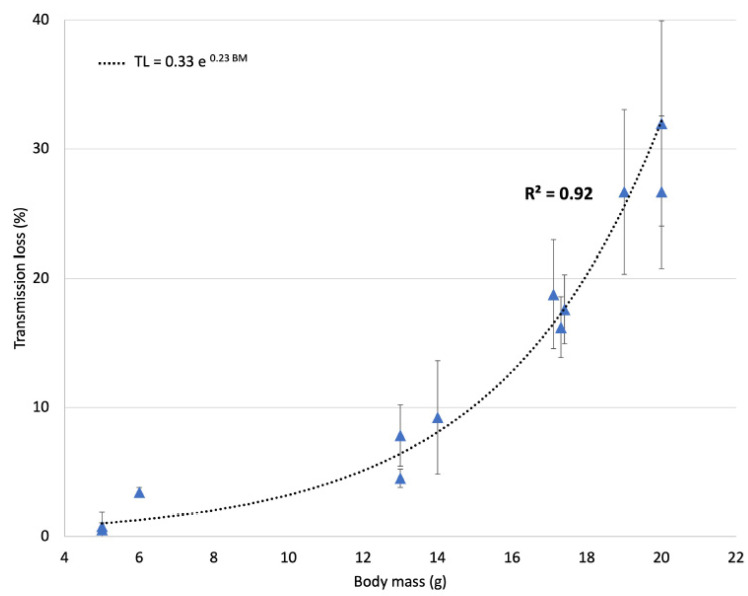
Ultrasound transmission loss through the mouse skull (*n* = 12) at 1.5 MHz as a function of the animal body mass. Each measurement was repeated 3 times.

**Figure 3 pharmaceutics-13-01245-f003:**
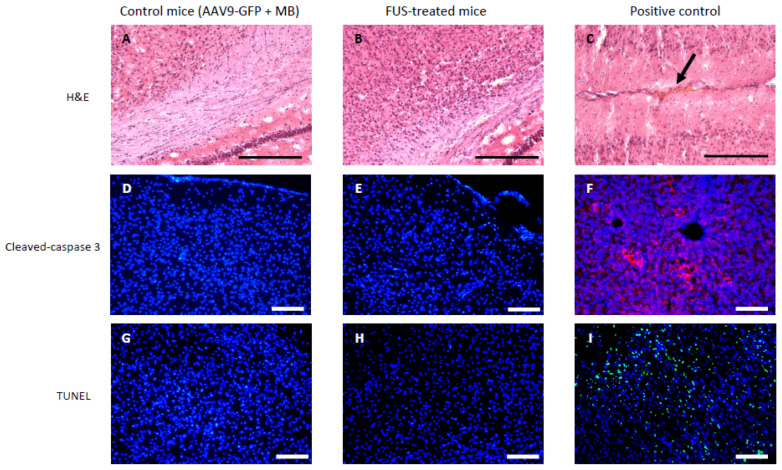
Safety studies 24 h after FUS treatment did not show any apoptosis or hemorrhages in mice brains. (**A**–**C**): Hematoxylin-eosin staining of mice that had FUS treatment after AAV9-GFP and microbubbles (MB) injections (**B**) showed no difference with control mice (only AAV9-GFP and MB injections) (**A**). Images are representative pictures at the corpus callosum and CA1 levels. Red blood cells are indicated with an arrow on the positive control image (**C**). (**D**–**F**): Cleaved-caspase 3 immuno-fluorescent red staining is absent in both control (**D**) and FUS-treated mice (**E**). Images are representative pictures in the striatum. Liver sections were used as a positive control (**F**). Cell nuclei are counterstained by DAPI staining (blue color). (**G**–**I**): TUNEL assay did not show signs of apoptotic cells in both control (**G**) and FUS-treated mice (**H**). Images are representative pictures showing TUNEL staining (green color), DNAse-treated brain sections were used as a positive control (**I**). Images are representative pictures in the striatum. Cell nuclei are counterstained by DAPI staining (blue color). The scale bar represents 100 μm (*n* = 3 for control mice and *n* = 3 for the FUS-treated mice).

**Figure 4 pharmaceutics-13-01245-f004:**
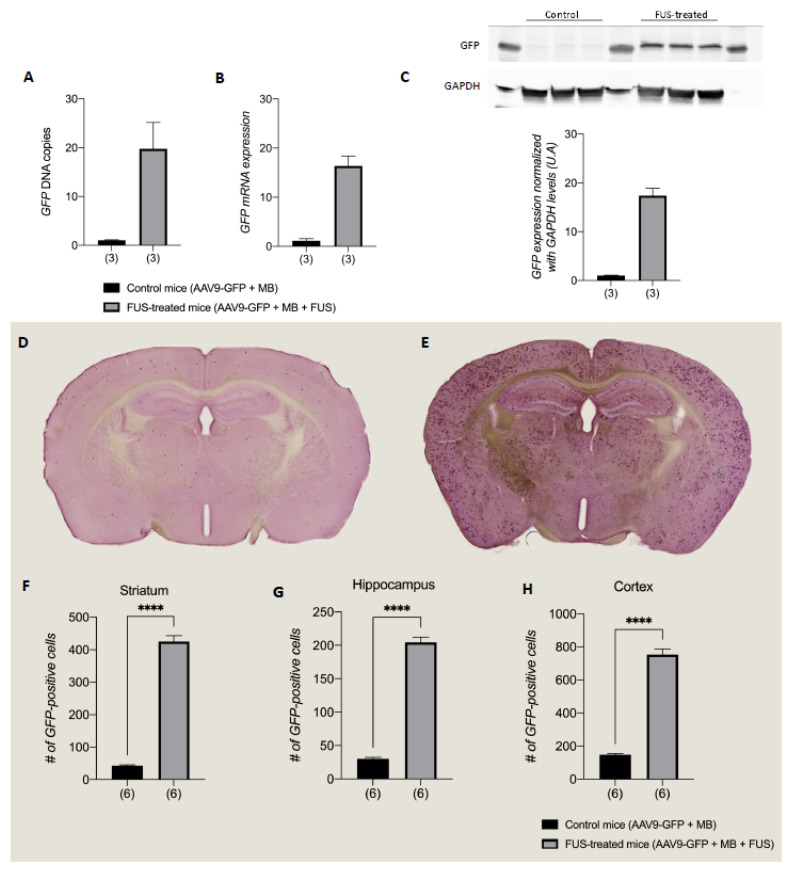
Efficiency studies 1 month after FUS treatment showed an increase of vector quantity, vector expression and number of infected cells in the brain. (**A**): FUS treatment increased AAV9-GFP transduction in the brain (cohen’s d effect size = 3.47). (**B**): GFP cDNA expression was increased in the brain of FUS-treated mice (cohen’s d effect size = 7.28). (**C**): Western blot quantification showed an increase in GFP expression in the brain after FUS treatment (cohen’s d effect size = 10.98). Representative western blot images with the GFP protein stained in black are shown above the graph. (**D**–**H**): Immunohistochemistry anti-GFP showed an increase in the number of GFP-positive cells in the brain of FUS-treated mice. Representative mages with the GFP protein stained in violet are shown above the graph for control mice (**D**) and FUS-treated mice (**E**). The number of GFP-positive cells was counted in the striatum (**F**), the hippocampus (**G**), and the cortex (**H**). Mean + SEM are represented by vertical bars, ****: *p* < 0.0001 Mann–Whitney rank-sum test, and (*n*) number of animals per group.

**Figure 5 pharmaceutics-13-01245-f005:**
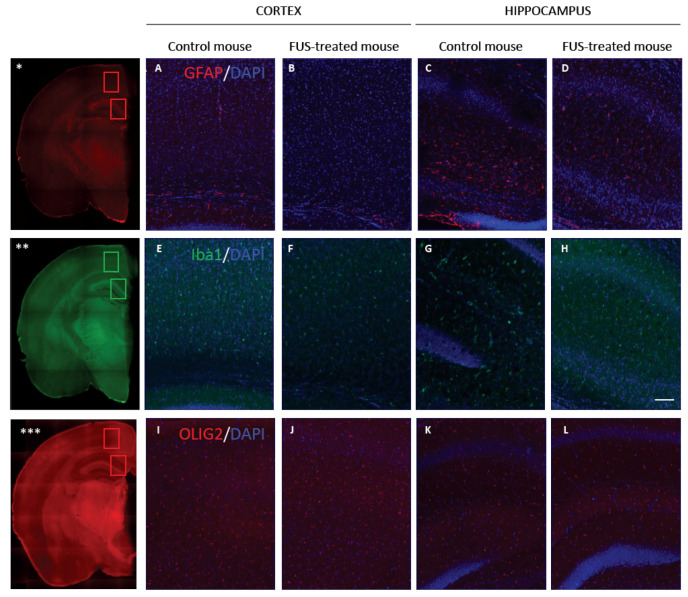
Long term safety study 1 month after FUS treatment did not show any inflammation, white matter damage or microglial activity. (**A**–**D**): GFAP immuno-fluorescent red staining in cortex (**A**,**B**) and in hippocampus (**C**,**D**) of mice that had FUS treatment after AAV9-GFP and MB injections (**B**,**D**) showed no difference with control mice (only AAV9-GFP and MB injections) (**A**,**C**). (**E**–**H**): Iba1 immuno-fluorescent green staining in cortex (**E**,**F**) and in hippocampus (**G**,**H**) of mice that had FUS treatment (**F,H**) showed no difference with control mice (**E**,**G**). (**I**–**L)**: Olig2 immuno-fluorescent red staining in cortex (**I**,**J**) and in hippocampus (**K**,**L**) of mice that had FUS treatment (**J**,**L**) showed no difference with control mice (**I**,**K**). Cell nuclei are counterstained by DAPI staining (blue color). The scale bar represents 100 μm (*n* = 3 for control mice and *n* = 3 for the FUS-treated mice). Images *, **, and *** are low magnification of half slide of the brain stained in GFAP (*), Iba1 (**), and Olig2 (***) in order to localize where the big magnification have been taken.

**Table 1 pharmaceutics-13-01245-t001:** Summary table of the number of animals studied per group, experimental conditions, latency between FUS-induced BBB disruption and the sacrifice of the animal, and experiments performed on their brain.

		Group	*n*	AAV9-GFP + MB	FUS	Experiments
Safety	Short term (24 h)	FUS-treated mice	3	✔	✔	H&E staining TUNEL staining assay Cleaved caspase-3 Immunohistofluorescence
Control mice	3	✔	
Long term (1 month)	FUS-treated mice	3	✔	✔	Iba1 immunohistofluorescence GFAP immunohistofluorescence Olig2 immunohistofluorescence
Control mice	3	✔	
Efficacy	Long term (1 month)	FUS-treated mice	3	✔	✔	GFP DNA quantification (qPCR) GFP mRNA quantification (RT-qPCR) GFP protein quantification (WB)
Control mice	3	✔	
FUS-treated mice	6	✔	✔	GFP immunohistochemistry Positive cells counted
Control mice	6	✔

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
