# Peer review of "Ultrasound-Mediated Blood-Brain Barrier Opening Improves Whole Brain Gene Delivery in Mice"

_pharmaceutics, 2021, doi:10.3390/pharmaceutics13081245_

Round 1

Reviewer 1 Report

The Authors present a sound experiment of modest significance - providing evidence for the use of FUS to deliver AAVs to the whole nervous system (as opposed to specific brain regions), as applied in a mouse model. The paper is a proof of concept that does open further applications. Some grammatical restructuring is recommended, along with clarification/additional proof of specific results. 

Recommendations:

  1. Please cite "The CNS is among the most targeted organs"
  2. In the introduction "stable cavitation" should be defined and outlined prior to its first mention.
  3. Immunofluorescent images demonstrating cell density/activation/and inflammation need fluorescent quantification. Claims can not be based on representative pictures alone. Figure 4 and Figure 6 require quantification data. Claims in conclusions can not be supported based only on representative images. 
  4. To adequately demonstrate that all glial cells are unhampered, an additional stain for oligodendrocytes (or white matter) should be considered. White matter damage can occur apart from apoptosis or DNA damage.  

Author Response

The Authors present a sound experiment of modest significance - providing evidence for the use of FUS to deliver AAVs to the whole nervous system (as opposed to specific brain regions), as applied in a mouse model. The paper is a proof of concept that does open further applications. Some grammatical restructuring is recommended, along with clarification/additional proof of specific results. 

Recommendations:

1 Please cite "The CNS is among the most targeted organs"

The citation has been added (page 2, line 4).

2 In the introduction "stable cavitation" should be defined and outlined prior to its first mention.

Stable cavitation has been defined in the introduction: “These gas-filled 1-5 mm MB can repeatedly expand and compress with no risk of collapsing provided when they are exposed to low enough acoustic pressures. This oscillation regime also called stable cavitation oscillating mode leads to tight junctions opening” (page 2, lines 16-19).

3 Immunofluorescent images demonstrating cell density/activation/and inflammation need fluorescent quantification. Claims can not be based on representative pictures alone. Figure 4 and Figure 6 require quantification data. Claims in conclusions can not be supported based only on representative images. 

Quantifications were done for the Figure 6 (Figure 5 in the revised manuscript) and no significant differences were found between FUS-treated and control mice. Data are presented in the Supplementary Figure 2. Unfortunately, for the Figure 4 (Figure 3 in the revised manuscript), no staining was obtained for cleaved-caspase 3 and TUNEL therefore quantifications could not be performed.

4 To adequately demonstrate that all glial cells are unhampered, an additional stain for oligodendrocytes (or white matter) should be considered. White matter damage can occur apart from apoptosis or DNA damage.  

Immunostaining anti-Olig2 (oligodendrocytes) and quantification were performed, and no white matter damage were observed. Data are presented in the Figure 5 and Supplementary Figure 2.

Reviewer 2 Report

In this article, the authors proposed focused ultrasound (FUS) coupled with microbubbles as the BBB opening device and implemented it to deliver a self-complementary AAV9-CMV-GFP vector into mice brain. This manuscript is well-written and explains the research clearly. The studies were described in details and the safety of FUS has been evaluated. Although using FUS for brain delivery is not at greatest novelty, this study identified the gap for whole-brain BBB-opening and can be a potentially contribution to the field. Overall, the manuscript is suitable for publication in its current form.

Author Response

In this article, the authors proposed focused ultrasound (FUS) coupled with microbubbles as the BBB opening device and implemented it to deliver a self-complementary AAV9-CMV-GFP vector into mice brain. This manuscript is well-written and explains the research clearly. The studies were described in details and the safety of FUS has been evaluated. Although using FUS for brain delivery is not at greatest novelty, this study identified the gap for whole-brain BBB-opening and can be a potentially contribution to the field. Overall, the manuscript is suitable for publication in its current form.

We thank the reviewer 2 for his/her comments.

Reviewer 3 Report

Dear authors,

greetings for your manuscript. A great work that deepens in permeabilizing the entire brain for gene therapy to threat different cerebral pathologies (for example neurodegenerative diseases). 

Authors have used different laboratory techniques and neuromarkers in order to demonstrate that FUS technology does not induce major brain side effects.

Only two minimal observations in the abstract:

-but also one of the most complex organs to reach

-Our results show that ... Our results show that 

warm wishes

Author Response

Dear authors,

greetings for your manuscript. A great work that deepens in permeabilizing the entire brain for gene therapy to threat different cerebral pathologies (for example neurodegenerative diseases). 

Authors have used different laboratory techniques and neuromarkers in order to demonstrate that FUS technology does not induce major brain side effects.

Only two minimal observations in the abstract:

-but also one of the most complex organ to reach

-Our results show that ... Our results show that 

warm wishes

We thank the reviewer 3 for his/her comments. The abstract was modified.

Reviewer 4 Report

In the article ‘ultrasound mediated blood brain barrier opening improves whole brain gene delivery in mice’, authors have shown that ultrasound sonication on whole brain area after AAV2/0 viral vector and SonoVue® injection with raster scanning device can improve the transfection effect up to almost 20 times. This article contains intriguing information since transfection a large part of brain thru BBB is difficult to achieve in controllable manner. However, there are many concerning matters in the current presentation.

  1. Premature conclusion is presented. Authors claim that TL through skull is a function of body mass based on 12 data at a single frequency result. I believe the data is as it is. And I totally agree that authors can refer the data to adjust for the successive experiments as guidance. But it is not logical to simplify the relation as such. Especially, the data shows that the samples not evenly. In addition, if there is relationship then authors need to provide physical reasoning to support that. The writing seems to lead to misreading.
  2. There are lack of consistency in writing material and equipment. Some of them contains all the information but others are missing. For example, ‘preserved in PBS azide’ in page 4 and ‘kept at 4℃ at PBS azide (0.02 % …)’. Another example, oscilloscope (MSO54, Tektronix, Beaverton, OR, USA) and Shandon Immu0Mount (ThermFisher Scientific). Author have to check the writing before submission.
  3. Composition of presentation is not well structured. For example, Figure 1 A and Figure 2 does not contain much of information for readers. Authors need to choose the presentation material carefully for readers.

Hence, this presentation requires major rewriting before publication

Comment in detail.

Abstract

  1. In abstract, there is not method which separate this presentation with others. As author suggested in introduction, there has been researches to transfect Gene through BBB with ultrasound and microbubbles. This article suggested that authors targeted for a whole brain. According to context, raster scan method was used to expose whole brain to ultrasound field. That has to be clarified.

Introduction

  1. ‘a highly selective permeable …‘. Authors may want to say ‘‘a highly selectively permeable …‘

  1. ‘Gene therapy approaches respresent …’ this sentence seems to be a starting of new paragraph.

  1. From the end of 1st page to the begging of 2nd page, four sentences are not well put together. Authors need to rewrite this part.

  1. In the last sentence, ‘to developer’. Did author intend to write ‘to develop’

Material and methods

  1. The number animal used are 30 which is not small number. However, if we see the detail, each group has 3 subjects (6 in last two). I understand there will be numerous image data and cell data. Hence, authors may think the small number of cases will be reasonable. But as a reviewer, I am a bit concerned. Can authors provide rational for this question?

  1. ‘All measures were taken to minimize animal suffering’ This sentence needs to be removed. If the experiment is conducted according to the institute regulation, then it has to be considered already. It seems meaningless redundancy.

  1. Figure 1. A does not provide much of help to reader. And the figure itself also a design picture. Please, choose wisely for the presentation. Figure 1. B should be moved to FUS0incuded BBB disruption part.

  1. Focused ultrasound device should only describe ultrasound system and calibration. Hence, from the last sentence of page 3 to the end seem to fit more ‘FUS-induced BBB disruption’ part. Scanning device is not part of Focused ultrasound device.

0.65 MPa appears in number of time in context. But it some cases it was definitely described as a peak negative pressure value. I believe the use of peak negative value related to Mechanical Index since it is an indirect indicator of inertia cavitation. If then, I believe authors need to explain it. If not, still we need some explanation.

  1. In ultrasound transmission through the skull. 12 mice were used but the weight distribution not even. For authors, it is necessary to describe, such as 5~10g :n1, 10~15g: n2, … and so on.

  1. PBS azide: the solution % was not given.

  1. ‘the single element transducer (Imasonic)’. There is not details of company information which is give some times. There are other similar cases in the context but I am not going to point out any further.

  1. ‘FUS-induced BBB disruption’

As pointed out in the previous section, ultrasound sonication part needs to be moved in this section. In addition, intensity of the sonication needs to be provided. Since continuous wave was used Ispta and Isppa will be same. Intensity level will help the readers how much energy is used directly. As a reviewer, I am also a bit concerned about overheating on skin surface. Even though intensity ultrasound on skin is much smaller than at the focus, still long exposure may cause harmful effects.

  1. ‘Tissue collection’ section

‘(1) for H&E stating ...’ the letter ‘f’ needs to be changed to ‘F’

‘(2) for Iba1 ...’ the letter ‘f’ needs to be changed to ‘F’

‘(3) for GFPquantification ...’ the letter ‘f’ needs to be changed to ‘F’

Detail information of procedure (3) is missing. I believe brain was extracted according to a certain procedure.

  1. GFP immunohistochemistry.

‘A region of interest (ROI) was drawn’, but we do not know what is the criterion of that. Please, provide the criteria to define ROI

Result

  1. Transmission loss

Transmission loss through skull is more directly related to skull thickness and the composition of the skull not the weight of subject. I suspect that larger mouse may be more mature and has thicker skull. However, if the skull maturity reaches a certain level, the skull characteristics may not vary much with respect to subject weight. Unless authors have a good rational to support the given relationship, this part could lead to misreading to general readers

  1. Figure 4. ‘a* on the positive control image’ where is a*?

  1. Figure 5. What is **** indicating?

Discussion

  1. (data not shown). I hope authors include ‘not shown data’ in supplementary. It is not proper to claim a certain fact without evidence in scientific paper.

Author Response

In the article ‘ultrasound mediated blood brain barrier opening improves whole brain gene delivery in mice’, authors have shown that ultrasound sonication on whole brain area after AAV2/0 viral vector and SonoVue® injection with raster scanning device can improve the transfection effect up to almost 20 times. This article contains intriguing information since transfection a large part of brain thru BBB is difficult to achieve in controllable manner. However, there are many concerning matters in the current presentation.Premature conclusion is presented. Authors claim that TL through skull is a function of body mass based on 12 data at a single frequency result. I believe the data is as it is. And I totally agree that authors can refer the data to adjust for the successive experiments as guidance. But it is not logical to simplify the relation as such. Especially, the data shows that the samples not evenly. In addition, if there is relationship then authors need to provide physical reasoning to support that. The writing seems to lead to misreading.

1 There are lack of consistency in writing material and equipment. Some of them contains all the information but others are missing. For example, ‘preserved in PBS azide’ in page 4 and ‘kept at 4℃ at PBS azide (0.02 % …)’. Another example, oscilloscope (MSO54, Tektronix, Beaverton, OR, USA) and Shandon Immu0Mount (ThermFisher Scientific). Author have to check the writing before submission.

As suggested by the reviewer, the description of material and methods section was improved.

2 Composition of presentation is not well structured. For example, Figure 1 A and Figure 2 does not contain much of information for readers. Authors need to choose the presentation material carefully for readers.

In agreement with reviewer’s comment, Figure 1 was modified, and uninformative figures were removed.

3 Hence, this presentation requires major rewriting before publication

Comment in detail.

Abstract

In abstract, there is not method which separate this presentation with others. As author suggested in introduction, there has been researches to transfect Gene through BBB with ultrasound and microbubbles. This article suggested that authors targeted for a whole brain. According to context, raster scan method was used to expose whole brain to ultrasound field. That has to be clarified.

This has been clarified: “Our results show that, after i.v. administration and FUS exposure following a raster scan …” (page 1, lines 31-32).

4 Introduction

‘a highly selective permeable …‘. Authors may want to say ‘‘a highly selectively permeable

This has been corrected (Page 1 line 47).

‘Gene therapy approaches respresent …’ this sentence seems to be a starting of new paragraph.

This has been corrected (Page 2 line 1).

6 From the end of 1st page to the begging of 2nd page, four sentences are not well put together. Authors need to rewrite this part.

This part has been rewritten.

7 In the last sentence, ‘to developer’. Did author intend to write ‘to develop’

This has been corrected (page 2, line 30).

Material and methods

8 The number animal used are 30 which is not small number. However, if we see the detail, each group has 3 subjects (6 in last two). I understand there will be numerous image data and cell data. Hence, authors may think the small number of cases will be reasonable. But as a reviewer, I am a bit concerned. Can authors provide rational for this question?

We agree that some groups are small. Nevertheless, within each group the values are extremely homogeneous with very little variability. In this context, it did not seem necessary to sacrify more animals and in the time available it was impossible to complete the groups knowing that AAVs take 3 weeks to express their transgene.

9 ‘All measures were taken to minimize animal suffering’ This sentence needs to be removed. If the experiment is conducted according to the institute regulation, then it has to be considered already. It seems meaningless redundancy.

This sentence was removed.

10 Figure 1. A does not provide much of help to reader. And the figure itself also a design picture. Please, choose wisely for the presentation. Figure 1. B should be moved to FUS0incuded BBB disruption part.

Figure 1 was modified to answer reviewer 4’s concerns and uninformative figures were deleted.

11 Focused ultrasound device should only describe ultrasound system and calibration. Hence, from the last sentence of page 3 to the end seem to fit more ‘FUS-induced BBB disruption’ part. Scanning device is not part of Focused ultrasound device.

This has been modified.

12 0.65 MPa appears in number of time in context. But it some cases it was definitely described as a peak negative pressure value. I believe the use of peak negative value related to Mechanical Index since it is an indirect indicator of inertia cavitation. If then, I believe authors need to explain it. If not, still we need some explanation.

We would like to thank the reviewer for the relevant comment. All the pressures indicated in the manuscript are peak negative pressure (PNP). The reviewer is correct, we usually use the PNP as an indicator of inertial cavitation and bubble expansion. PNP is now clearly indicated into the manuscript. 

 13 In ultrasound transmission through the skull. 12 mice were used but the weight distribution not even. For authors, it is necessary to describe, such as 5~10g :n1, 10~15g: n2, … and so on.

This has been added (page 4, line 2)

14 PBS azide: the solution % was not given.

This has been corrected

15 ‘the single element transducer (Imasonic)’. There is not details of company information which is give some times. There are other similar cases in the context but I am not going to point out any further.

This has been corrected.

16 ‘FUS-induced BBB disruption’

As pointed out in the previous section, ultrasound sonication part needs to be moved in this section. In addition, intensity of the sonication needs to be provided. Since continuous wave was used Ispta and Isppa will be same. Intensity level will help the readers how much energy is used directly. As a reviewer, I am also a bit concerned about overheating on skin surface. Even though intensity ultrasound on skin is much smaller than at the focus, still long exposure may cause harmful effects.

We thank the reviewer for his/her comment. The safety and the efficiency of the raster scan protocol was already validated on rodents in a previous publication referenced into the manuscript (see ref 31). The reviewer must consider that the transducer is moving during the experiment. Concretely, the focal spot of the circular transducer has a 1 mm width and it is moving on the head of the animal at a velocity of 10 mm.s-1. It means that each spot along the trajectory is insonified for 0.1s. The raster scan is repeated 30 times every 5s. Therefore, the effective duty cycle for a targeted area, including the skin surface, is in the order of 2%. We should mention that no skin burn was observed using this protocol.

A sentence has been to the discussion section to clarify this point (page 9, lines 42-44).

‘Tissue collection’ section

17 ‘(1) for H&E stating ...’ the letter ‘f’ needs to be changed to ‘F’

18 ‘(2) for Iba1 ...’ the letter ‘f’ needs to be changed to ‘F’

19 ‘(3) for GFPquantification ...’ the letter ‘f’ needs to be changed to ‘F’

This has been corrected

20 Detail information of procedure (3) is missing. I believe brain was extracted according to a certain procedure.

Brains were quickly dissected and frozen as specified in procedure 3.

GFP immunohistochemistry.

21 ‘A region of interest (ROI) was drawn’, but we do not know what is the criterion of that. Please, provide the criteria to define ROI

Region of interest were brain structures (Cortex, striatum and hippocampus) and were defined according to the Allen Mouse Brain Atlas (2004). This has been specified in the material and methods section.

Result

21 Transmission loss

Transmission loss through skull is more directly related to skull thickness and the composition of the skull not the weight of subject. I suspect that larger mouse may be more mature and has thicker skull. However, if the skull maturity reaches a certain level, the skull characteristics may not vary much with respect to subject weight. Unless authors have a good rational to support the given relationship, this part could lead to misreading to general readers.

We totally agree that transmission loss is related to skull thickness and skull composition. However, contrary to the body mass, this information is not easily accessible on a mouse. Even the resolution of the CT scan would not be high enough to determine thickness and composition. For this reason, we were looking for an indicator, easy to measure, related to the skull maturation that would be used to estimate transmission loss and adjust the acoustic pressure. Transmission loss through the skull was successfully related to the body mass on rats (see ref. 34). Here, we demonstrated a relationship between the body mass and the transmission loss on young mice. We support that this relation is relevant. As already mentioned in the discussion section, this correction by the transmission loss is particularly necessary for animal models including young mice or pathology associated with growth retardation such as for Rett syndrome. Indeed, mice having a severe growth retardation have a body mass much lower than healthy mice. Our preliminary results (not shown into this manuscript) on diseased animals confirm that body mass is a much better indicator than age to estimate the transmission loss through the skull.

22 Figure 4. ‘a* on the positive control image’ where is a*?

The size of * has been enlarged to clarify the Figure.

23 Figure 5. What is **** indicating?

The **** indicates a p-value < at 0.0001

Discussion

25 (data not shown). I hope authors include ‘not shown data’ in supplementary. It is not proper to claim a certain fact without evidence in scientific paper.

The supplementary figure 3 was created to show brain hemorrhages.

Round 2

Reviewer 1 Report

The authors have sufficiently addressed previous concerns and added appropriate quantification figures. 

Author Response

We thank the reviewer 1 for his/her comments.

Reviewer 4 Report

I appreciate authors’ effort to clarify my concern.  This revision is immensely better than the first manuscript. This seems almost good enough for publication except for minor errors.

  1. Abstract

‘FUS exposure following a raster scan’. This expression seems not appropriate. Isn’t it ‘ultrasound sonincated in a raster scan manner’?

  1. Introduction

‘These gas-filled 1-5 m MB can repeatedly expand and compress with no risk of collapsing provided when they are exposed to low enough acoustic pressures.’ Is the word ‘provided’ is right here? And I believe it is supposed to be μm.

  1. Materials & Methods

Table 1: resolution is not good enough.

‘An in vivo platform (Image Guided Therapy, France) (Figure 2) comprises a stereotaxic frame,’ I believe in vivo should be italic.

‘5 g to 20 g, 5~10g: n3, 10~15g: n3, 15~20g: n6)’ There are cases in this article as shown here that no space between number and unit. ’10 g’

‘4°C’  -> 4 °C

‘The calibrated sonication parameters allowing BBB opening were fixed in order to deliver an in situ peak negative pressure of 0.57 MPa at the focus point.’ If PNP is used in the previous paragraph, you may want to keep use the term.

  1. Result

Figure 3. caption.  I am still having difficulty to find ‘*’ mark in figures. I recommend to change the mark.

Author Response

I appreciate authors’ effort to clarify my concern.  This revision is immensely better than the first manuscript. This seems almost good enough for publication except for minor errors.

We thank the reviewer 4 for his/her comments.

  1. Abstract

‘FUS exposure following a raster scan’. This expression seems not appropriate. Isn’t it ‘ultrasound sonincated in a raster scan manner’?

This has been corrected (Page 1 lines 31-32).

  1.  Introduction

‘These gas-filled 1-5 m MB can repeatedly expand and compress with no risk of collapsing provided when they are exposed to low enough acoustic pressures.’ Is the word ‘provided’ is right here? And I believe it is supposed to be μm.

The word ‘provided’ has been deleted (Page 2 line 16).

  1. Materials & Methods

Table 1: resolution is not good enough.

The Table 1 was modified to increase the resolution.

‘An in vivo platform (Image Guided Therapy, France) (Figure 2) comprises a stereotaxic frame,’ I believe in vivo should be italic.

 This has been corrected (Page 3 line 35).

‘5 g to 20 g, 5~10g: n3, 10~15g: n3, 15~20g: n6)’ There are cases in this article as shown here that no space between number and unit. ’10 g’

  This has been corrected (Page 4 lines 2-3).

‘4°C’  -> 4 °C

   A space was added wherever necessary.

‘The calibrated sonication parameters allowing BBB opening were fixed in order to deliver an in situ peak negative pressure of 0.57 MPa at the focus point.’ If PNP is used in the previous paragraph, you may want to keep use the term.

  This has been corrected (Page 5 line 3).

  1. Result

Figure 3. caption.  I am still having difficulty to find ‘*’ mark in figures. I recommend to change the mark.

The mark has been changed to an arrow to improve readability.